# Endogenous RNA interference is driven by copy number

Cristina Cruz, Jonathan Houseley*

Epigenetics Programme, The Babraham Institute, Cambridge, United Kingdom

**Abstract** A plethora of non-protein coding RNAs are produced throughout eukaryotic genomes, many of which are transcribed antisense to protein-coding genes and could potentially instigate RNA interference (RNAi) responses. Here we have used a synthetic RNAi system to show that gene copy number is a key factor controlling RNAi for transcripts from endogenous loci, since transcripts from multi-copy loci form double stranded RNA more efficiently than transcripts from equivalently expressed single-copy loci. Selectivity towards transcripts from high-copy DNA is therefore an emergent property of a minimal RNAi system. The ability of RNAi to selectively degrade transcripts from high-copy loci would allow suppression of newly emerging transposable elements, but such a surveillance system requires transcription. We show that low-level genome-wide pervasive transcription is sufficient to instigate RNAi, and propose that pervasive transcription is part of a defense mechanism capable of directing a sequence-independent RNAi response against transposable elements amplifying within the genome.

## Introduction

Over the past decade, our understanding of the complexity of the eukaryotic transcriptome has been revolutionized. Genome-wide sequencing studies in many organisms have revealed that protein-coding mRNAs are augmented by a multitude of non-protein coding RNAs (ncRNAs), many produced from regions of the genome traditionally considered to be transcriptionally silent (*The ENCODE Project Consortium, 2012*; *Bertone et al., 2004*; *Cheng et al., 2005*; *David et al., 2006*; *Birney et al., 2007*). Functional data for the vast majority of ncRNAs are currently lacking, with only a few examples characterized in any detail; however, the diversity of mechanisms by which these act suggests that ncRNAs have a rich and varied biology that is largely still to be sampled.

Long ncRNAs which overlap protein-coding genes have the potential to modulate the expression of their cognate coding RNA. Early characterized examples in yeast were thought to work by directly disrupting transcription factor or polymerase binding to the promoter of the coding RNA (*Martens et al., 2004*; *Hongay et al., 2006*); however, more recent data implicate specific chromatin structure changes in repression (*Gelfand et al., 2011*; *Hainer et al., 2011*), and many other cases of ncRNAs that alter chromatin modifications have been described (*Camblong et al., 2007*; *Berretta et al., 2008*; *Houseley et al., 2008*; *Pinskaya et al., 2009*; *van Werven et al., 2012*). Chromatin modifications are not necessarily repressive, and ncRNAs that enhance expression of their overlapping coding gene have also been described (*Uhler et al., 2007*; *Hirota et al., 2008*). In these examples, chromatin modifications are deposited during transcription, and therefore the act of transcription rather than the ncRNA itself is important. This is not always the case, and in higher eukaryotes multiple *cis*-acting ncRNAs have also been characterized, particularly as functional agents in imprinting. For example, *Air* and *Kcnq1ot1* act in *cis* to deposit repressive chromatin marks and DNA methylation, but these ncRNAs interact with chromatin modifiers and allele specificity is achieved by restriction of the ncRNA to the vicinity of the transcription site, although the importance of the transcription itself remains controversial (*Nagano et al., 2008*; *Pandey et al., 2008*; *Redrup et al., 2009*; *Latos et al., 2012*).

*For correspondence: jon. houseley@babraham.ac.uk

Competing interests: The authors declare that no competing interests exist.

**eLife digest** Genes contain the codes that are needed to make the proteins used by cells. This code is transcribed to make a messenger RNA molecule that is then translated to make a protein. However, other types of RNA called non-coding RNA molecules can disrupt this process by binding to messenger RNA molecules, with matching sequences, before translation begins. This phenomenon, which is known as RNA interference, involves enzymes called Dicer and Argonaute.

Many cells contain large numbers of non-coding RNA molecules—so called because they are not translated to produce proteins—and many of these are capable of starting the process of RNA interference. However, most do not, and the reasons for this are not understood. Now, work by Cruz and Houseley has provided new insight into this phenomenon by showing that it is related to the number of copies of the gene encoding such RNAs in the genome.

Yeast cells normally do not have the genes for RNA interference, but Cruz and Houseley used genetically engineered yeast cells containing Dicer and Argonaute. Although most of the messenger RNA molecules in these cells showed no change, the expression of some genes with high 'copy numbers' was reduced. Further experiments that involved adding more and more copies of other genes showed that RNA interference could selectively target messenger RNA molecules produced from genes with an increased copy number—particularly if the copies of the genes were clustered in one location in the genome.

RNA interference is also used to defend against DNA sequences that invade and multiply within a genome, such as viruses and other 'genetic parasites'. As such, the effect observed by Cruz and Houseley could explain why entire genomes are often continuously copied to RNA at low levels. This activity would allow the monitoring of the genome for the invasion of any genetic parasites that had multiplied to high numbers. Following on from this work, the next challenge will be to understand how gene copy number and location are balanced to achieve a selective RNA interference system.

Genomes are also replete with low abundance and unstable RNA. The vast majority of ncRNAs in budding yeast are unstable (*Neil et al., 2009*; *van Dijk et al., 2011*), limiting the potential action of the RNAs themselves, although the transcription of such RNAs can still alter gene expression (reviewed in *Houseley, 2012*). Such unstable RNAs are also widespread in higher eukaryotes, probably with similar functional roles (*Chekanova et al., 2007*; *Preker et al., 2008*). More mysterious is the pervasive transcription that permeates eukaryotic genomes; the ENCODE project found that almost all the human genome is transcribed at some point, but the products of this transcription are vanishingly rare (*Cheng et al., 2005*; *Birney et al., 2007*; *Kapranov et al., 2007*; *Goodman et al., 2012*). It appears that regions of the genome which are not actively transcribed for other reasons undergo pervasive transcription; however, it is not known whether this pervasive transcription simply represents transcriptional noise or whether the transcription or RNAs themselves have important but as yet undiscovered functions.

Systems in which a ncRNA is transcribed antisense to a sense protein-coding RNA are common and have strong regulatory potential (*Figure 1A*) (*Derrien et al., 2012*; *Carninci et al., 2005*; *Xu et al., 2009*). It has been suggested that, since antisense ncRNAs are perfectly complementary to their cognate mRNA, the two species could form double stranded RNA (dsRNA) that would be a substrate for the RNA interference system (RNAi). During a basic RNAi response, dsRNA is cleaved by the endonuclease Dicer into short interfering RNA (siRNA), of which one strand is then loaded onto an Argonaute protein. The Argonaute–siRNA complex can anneal to complementary sequences in target RNAs, which are then cleaved by the endonuclease activity of Argonaute. RNAi was originally discovered in *Caenorhabditis elegans* and rapidly linked to the phenomenon of post-transcriptional gene silencing in plants; however, almost all eukaryotes contain Dicer and Argonaute orthologues and therefore have some form of RNAi system (*Hamilton and Baulcombe, 1999*; *Fire et al., 1998*; *Hannon, 2002*). RNAi probably evolved to protect cells against dsRNA viruses, a role which is maintained in plants, insects, and lower eukaryotes (*Ding, 2010*) and has recently been described in mammalian cells (*Li et al., 2013*; *Maillard et al., 2013*). RNAi also forms a potent defense against transposons, and high-copy transposon-derived sequences are excellent targets for RNAi, giving rise to copious siRNAs in most eukaryotes including mammals (*Yang and Kazazian, 2006*; *Slotkin and Martienssen, 2007*; *Babiarz et al., 2008*).

In addition to degrading transposon-derived and viral RNA, siRNAs can mediate transcriptional repression of target RNAs through chromatin modifications and DNA methylation, although this activity is seemingly much stronger in lower eukaryotes and plants than in mammals (*Martienssen et al., 2005*; *Lejeune et al., 2010*; *Zhang and Zhu, 2011*). However, the source of the dsRNA that is processed into siRNA is not always obvious, nor is the mechanism by which cells differentiate host and transposon-derived sequences. siRNA-mediated repression is complemented by PIWI-interacting RNAs (piRNAs), which bind to the Argonaut-related PIWI-domain proteins and enforce transposon repression in the germline of eukaryotes from worms to mammals (*Siomi et al., 2011*). piRNAs are derived from specific genomic clusters, but it is unclear how the transcripts from these clusters are selected for processing into primary piRNAs and many of the processing enzymes remain to be identified (reviewed in *Ishizu et al., 2012*).

Various studies have looked for endogenous sense-antisense RNA pairs that instigate RNAi responses. Efficient generation of siRNAs from endogenous sense-antisense systems (endo-siRNA) has been observed in plants under stress (*Borsani et al., 2005*; *Katiyar-Agarwal et al., 2006*), and mammalian oocytes generate endo-siRNAs that can mediate mRNA knockdown (*Tam et al., 2008*; *Watanabe et al., 2008*). However, although endo-siRNAs have been detected outside the germline in mammals, they are

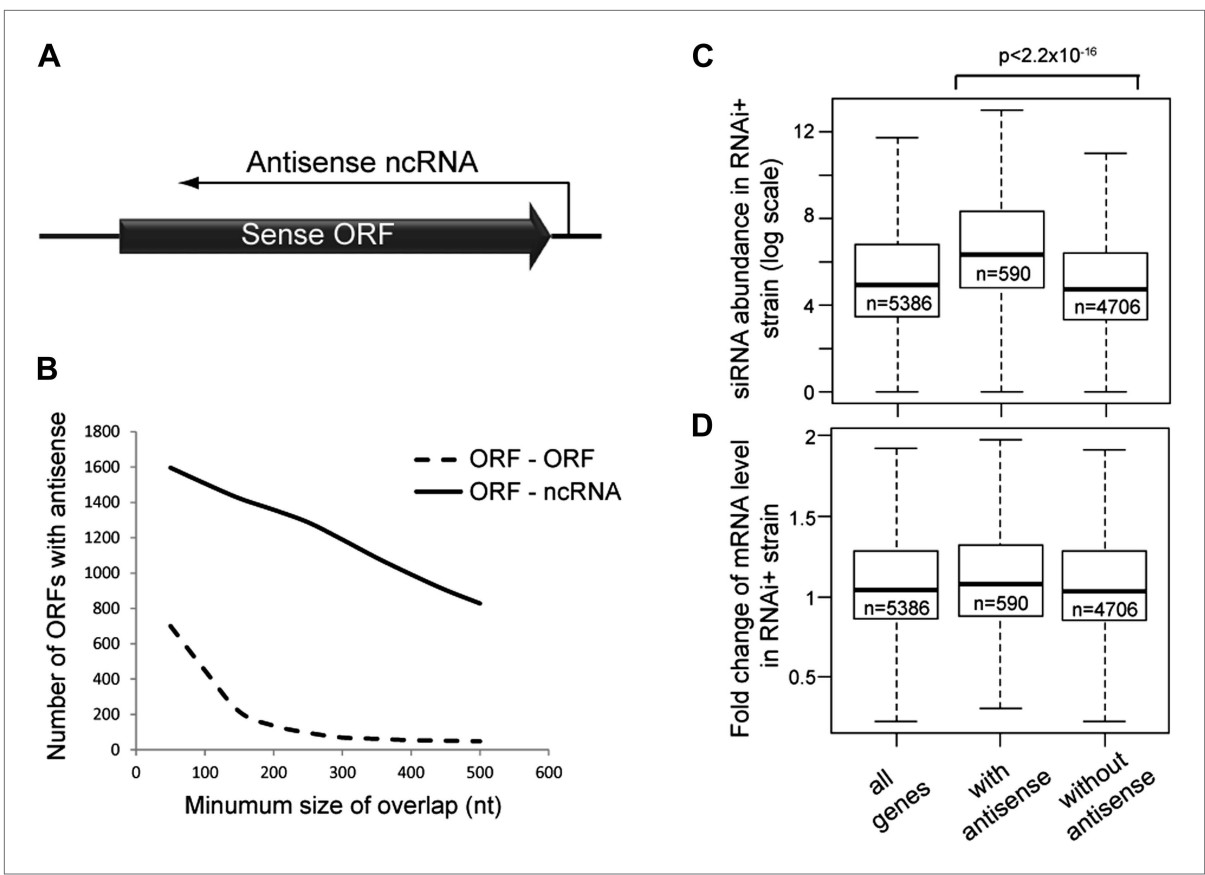

**Figure 1**. Frequency of annotated antisense non-protein coding RNAs (ncRNAs) and effects on mRNA abundance. (**A**) Schematic of an example sense mRNA-antisense (ncRNA) system. (**B**) Number of annotated open reading frames (ORFs) with antisense transcripts. Positions of CUTs, SUTs, and XUTs were collated with expressed ORFs (*Xu et al., 2009*; *van Dijk et al., 2011*), SUTs were later re-classified as XUTs were removed. Overlaps between ORFs expressed in glucose media (total 5171, *Xu et al., 2009*) and other RNAs were calculated and summed for increasing minimum overlaps of 50–500 bp. ORF–ORF overlaps and ORF–ncRNA overlaps were analyzed separately as ORF–ORF overlaps are consistently smaller. Detailed figures are given in *Table 1*. (**C**) Abundance of short interfering RNAs (siRNAs) in RNA interference (RNAi)+ strain produced from expressed ORFs with and without an annotated overlapping antisense ncRNA, based on read counts from published high-throughput sequencing data (*Drinnenberg et al., 2011*). Minimum antisense overlap with ORF was set at 250 bp; only ORFs with >100 reads in the wild-type poly(A)+ library were assessed to remove noise. Stated p value calculated by Student's *t* test. (**D**) Abundance of mRNA in RNAi+ cells relative to wild-type; data source and categories as in **C**, differences were not significant.

**Table 1.** Stability of antisense ncRNAs

| Overlap size (bp) | Overlap type | | | | Totals | |
|---|---|---|---|---|---|---|
| | ORF–ORF | ORF–XUT | ORF–CUT | ORF–SUT | ORF–ncRNA | ORF–unstable ncRNA |
| 50 | 700 | 1066 | 575 | 522 | 1596 | 1448 (91%) |
| 100 | 449 | 1008 | 543 | 475 | 1507 | 1367 (91%) |
| 150 | 216 | 967 | 508 | 425 | 1423 | 1306 (92%) |
| 200 | 136 | 931 | 478 | 403 | 1358 | 1249 (92%) |
| 250 | 96 | 893 | 434 | 380 | 1287 | 1181 (92%) |
| 300 | 69 | 812 | 391 | 363 | 1189 | 1085 (91%) |
| 350 | 62 | 759 | 335 | 351 | 1086 | 990 (91%) |
| 400 | 54 | 694 | 292 | 334 | 992 | 899 (91%) |
| 450 | 51 | 637 | 244 | 322 | 904 | 811 (90%) |
| 500 | 48 | 591 | 204 | 302 | 828 | 741 (89%) |

Number of ORFs overlapping with ORFs and various classes of ncRNAs, with various minimum size cut-offs for the overlapping region.
Totals are given for ORFs overlapping with ncRNAs and with unstable ncRNAs, including a percentage of overlapping ncRNAs that are unstable.
ncRNA: non-protein coding RNA; ORF: optical reading frame; XUT: Xrn1-sensitive unstable transcript (degraded in cytoplasm); CUT: cryptic unstable transcript (degraded by nuclear exosome); SUT: stable unannotated transcript (not known to be degraded).

surprisingly under-represented where sense and antisense RNAs are co-expressed (*Faghihi and Wahlestedt, 2006*; *Okamura et al., 2008*; *Carlile et al., 2009*), and overall there is a positive correlation between antisense and sense RNA expression in mammalian genomes, which is inconsistent with RNAi (*Derrien et al., 2012*; *Katayama et al., 2005*). This raises questions about whether endogenous sense-antisense systems do in fact form dsRNA in vivo and, if so, whether all dsRNA is equivalently accessible to Dicer.

The tight integration of the RNAi system into the physiology of most eukaryotic cells makes it very difficult to disentangle direct and indirect effects of mutating RNAi components (reviewed in *Ketting, 2011*). To elucidate factors important for the induction of RNAi by endogenous sense-antisense systems, we therefore used a recently described synthetic system in which RNAi is reconstituted in *Saccharomyces cerevisiae* by the introduction of Argonaute and Dicer from the related yeast *S. castellii* (*Drinnenberg et al., 2009*). *S. cerevisiae* is highly unusual in lacking an endogenous RNAi system, allowing maintenance of the symbiotic dsRNA Killer virus (*Drinnenberg et al., 2011*). The reconstituted system is functional, since RNAi+ *S. cerevisiae* efficiently degrades exogenous hairpin RNAs and endogenous Ty retrotransposon transcripts; however, no clear mRNA expression changes are detectable in these cells (*Drinnenberg et al., 2009*, *2011*).

## Results

Antisense ncRNAs exist for many *S. cerevisiae* genes; combining published datasets we found that 15–30% of yeast open reading frames (ORFs) have an annotated antisense ncRNA depending on the minimum size of overlap considered (*Xu et al., 2009*; *van Dijk et al., 2011*), not including overlapping convergent ORFs (*Figure 1B* and *Table 1*). In RNAi+ cells, these ORFs produce more siRNA than ORFs lacking an antisense (*Figure 1C*), showing that they are transcribed into dsRNA that is targeted by the RNAi machinery. However, these ORFs do not show reduced mRNA levels in RNAi+ cells consistent with published data, suggesting that insufficient siRNAs are produced to elicit a detectable mRNA knockdown (*Derrien et al., 2012*; *Katayama et al., 2005*; *Drinnenberg et al., 2011*) (*Figure 1D*).

We first asked whether any endo-siRNA pairs are degraded by RNAi in this reconstituted system. RNAi+ cells produce abundant siRNAs from sub-telomeric Y′ elements and from the ribosomal DNA (rDNA) intergenic spacers (*Drinnenberg et al., 2011* and *Figure 2—figure supplement 1*) and, despite transcriptional repression by the histone deacetylase Sir2, both regions transcribe sense and antisense

ncRNAs that could hybridize to form dsRNA (*Aparicio et al., 1991*; *Yamada et al., 1998*; *Kobayashi and Ganley, 2005*; *Houseley et al., 2007*) (*Figure 2A,D*). Northern blots revealed full-length ncRNAs from both strands of the Y′ elements in wild-type cells (*Figure 2B* lanes 1,5 marked with arrows); these were largely absent in the RNAi+ strain, being replaced by heterogeneous degradation products and readily detectable siRNAs (*Figure 2B,C*). Despite weak transcriptional repression in this genetic background, ncRNAs and siRNAs were more abundant in *sir2Δ* mutants reinforcing the precursor–product relationship (*Figure 2B,C*). Equivalent results were seen for the rDNA intergenic spacer region (*Figure 2D–F*). These data show that endo-siRNA pairs can form RNAi substrates and undergo efficient degradation by a minimal RNAi system.

One distinguishing feature of these regions is high copy number; to determine whether copy number amplification can drive RNAi, we examined *MAL32* which has a clearly defined antisense RNA that is co-expressed with the sense mRNA (*Figure 3—figure supplement 1*). *MAL32* is effectively present at two copies in the haploid genome as the orthologous gene *MAL12* shares 99.5% nucleotide identity, reducing potential transcriptional repression. As for approximately 90% of yeast antisense RNAs (*Table 1*), *MAL32* antisense RNA is highly unstable and is only detectable in strains lacking the nuclear exosome co-factor Trf4 (reviewed in *Houseley and Tollervey, 2009*) (*Figure 3—figure supplement 1*), but endogenous *MAL32* mRNA was not down-regulated in the RNAi+ strain even in *trf4Δ* cells (*Figure 3B* and *Figure 3—figure supplement 2A*). When expressed from a high-copy plasmid, however, *MAL32* mRNA was significantly down-regulated by RNAi (*Figure 3C*) with concurrent production of siRNA (*Figure 3D*). The *MAL32* antisense RNA was only detected in these experiments as a smear of degradation products and was not noticeably altered by RNAi, probably because nuclear degradation acts faster than RNAi on this substrate. To confirm that the knockdown of *MAL32* mRNA was not an indirect effect of the strain background or an undirected Argonaute cleavage, we reconstituted the RNAi system in the BY4741 background using separate plasmids expressing Dicer and Argonaute. dsRNA from the *MAL32* locus was detectable in these cells and was removed by Dicer; however, a significant knockdown of the mRNA was only observed in cells expressing both Dicer and Argonaute, confirming that the knockdown represents a genuine RNAi response (*Figure 3—figure supplement 3*).

We then tested *GAL4*, which is a single-copy gene with a co-expressed antisense that is degraded in the cytoplasm by Xrn1 (*Geisler et al., 2012*) (*Figure 3E*). Cells lacking Xrn1 show increased levels of antisense and, unexpectedly, sense RNA, but, as for *MAL32*, we did not detect a significant decrease in full-length RNA in RNAi+ *xrn1Δ* cells (*Figure 3F* and *Figure 3—figure supplement 2B*). However, amplification of the locus by cloning on a high-copy plasmid leads to significant degradation of the sense RNA along with the antisense RNA by RNAi (*Figure 3G,H*). Surprisingly, this occurred even in a known mutant that lacks antisense expression (*Geisler et al., 2012*), but 5′ RACE (Rapid Amplification of 5′ Complementary DNA Ends) experiments revealed that antisense RNA is still produced by this mutant when expressed from a high-copy plasmid, even if it is too heterogeneous for detection by northern blot (*Figure 3—figure supplement 4*). Taken together, these experiments on *MAL32* and *GAL4* demonstrate that increasing gene copy number can make the products of a normal gene susceptible to RNAi.

One potential confounding factor in these experiments is the copy number of the high-copy plasmids; if the copy number drops in RNAi+ cells, this would provide a trivial explanation for the observed knockdowns. However, Southern blotting revealed that RNAi+ cells contained approximately twofold more plasmid than the controls, which would tend to decrease rather than increase the apparent effect of RNAi (*Figure 3—figure supplement 5A,B*). It is likely that RNAi degrades the mRNA for the plasmid-encoded selectable marker and 2μ maintenance genes (2μ genes produce copious siRNA, *Figure 3—figure supplement 5C*), and the plasmid copy number rises to compensate for this. We also wanted to use a different method to confirm the northern blot results. We therefore lysed wild-type and RNAi+ cells containing the *MAL32* and *GAL4* plasmids, precipitated dsRNA using a specific monoclonal antibody (*Schonborn et al., 1991*; *Gullerova and Proudfoot, 2012*), and assayed total and dsRNA levels by quantitative RT-PCR. *MAL32* mRNA was knocked down approximately 75%, as observed by northern blot analysis, while dsRNA was reduced 11-fold, consistent with specific removal of dsRNA by Dicer. *GAL4* mRNA knockdown was measured at 80% in this assay, again with an 11-fold reduction in dsRNA in the RNAi+ strain (*Figure 3—figure supplement 6*).

Increasing gene copy number also increases RNA production. To separate the contributions of RNA abundance and copy number, we analyzed existing genome-wide data (*Hobson et al., 2012*; *Drinnenberg et al., 2011*). If siRNA formation depends only on precursor RNA abundance, a positive correlation

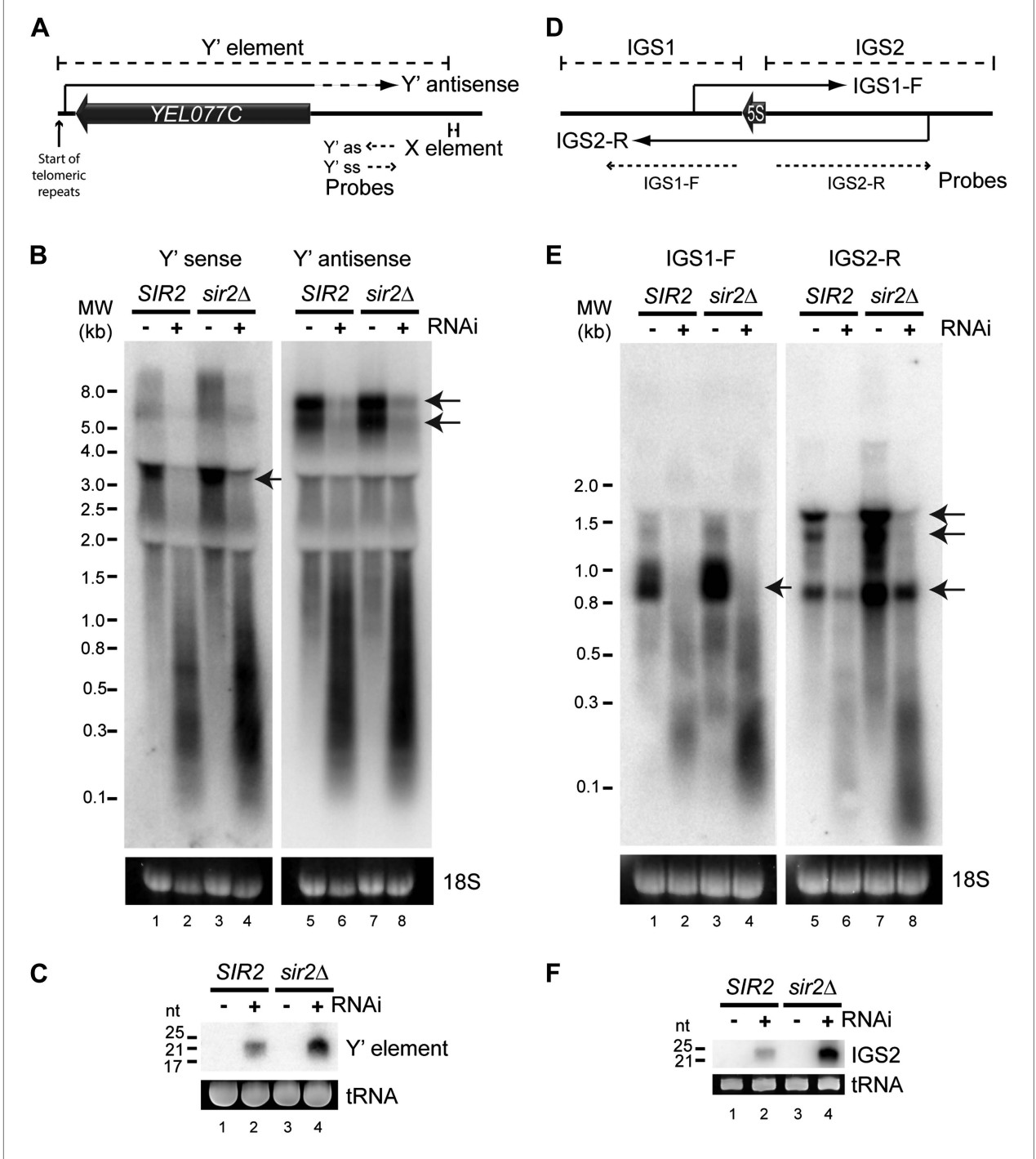

**Figure 2**. High-copy endogenous sense-antisense pairs instigate efficient RNA interference (RNAi). (**A**) Schematic diagram of sub-telomeric Y' elements. (**B**) Northern analysis of Y' element non-protein coding RNAs (ncRNAs) comparing wild-type and RNAi+ strains in *SIR2* and *sir2*Δ backgrounds. 18S ribosomal RNA is shown as a loading control. Arrows indicate full-length RNA species. (**C**) Northern analysis of Y' element-derived short interfering RNAs (siRNAs) from cells in **B**, tRNAs are shown as a loading control. (**D–F**) Equivalent analysis of rDNA intergenic spacer ncRNAs.

The following figure supplements are available for figure 2:

**Figure supplement 1**. Correlation of short interfering RNAs (siRNAs) with RNA abundance and silencing. Published small RNA sequencing data (**Drinnenberg et al., 2011**) were re-mapped to the complete *Saccharomyces cerevisiae* genome including non-unique sequences and reads were then summed in 50 bp bins on each strand.

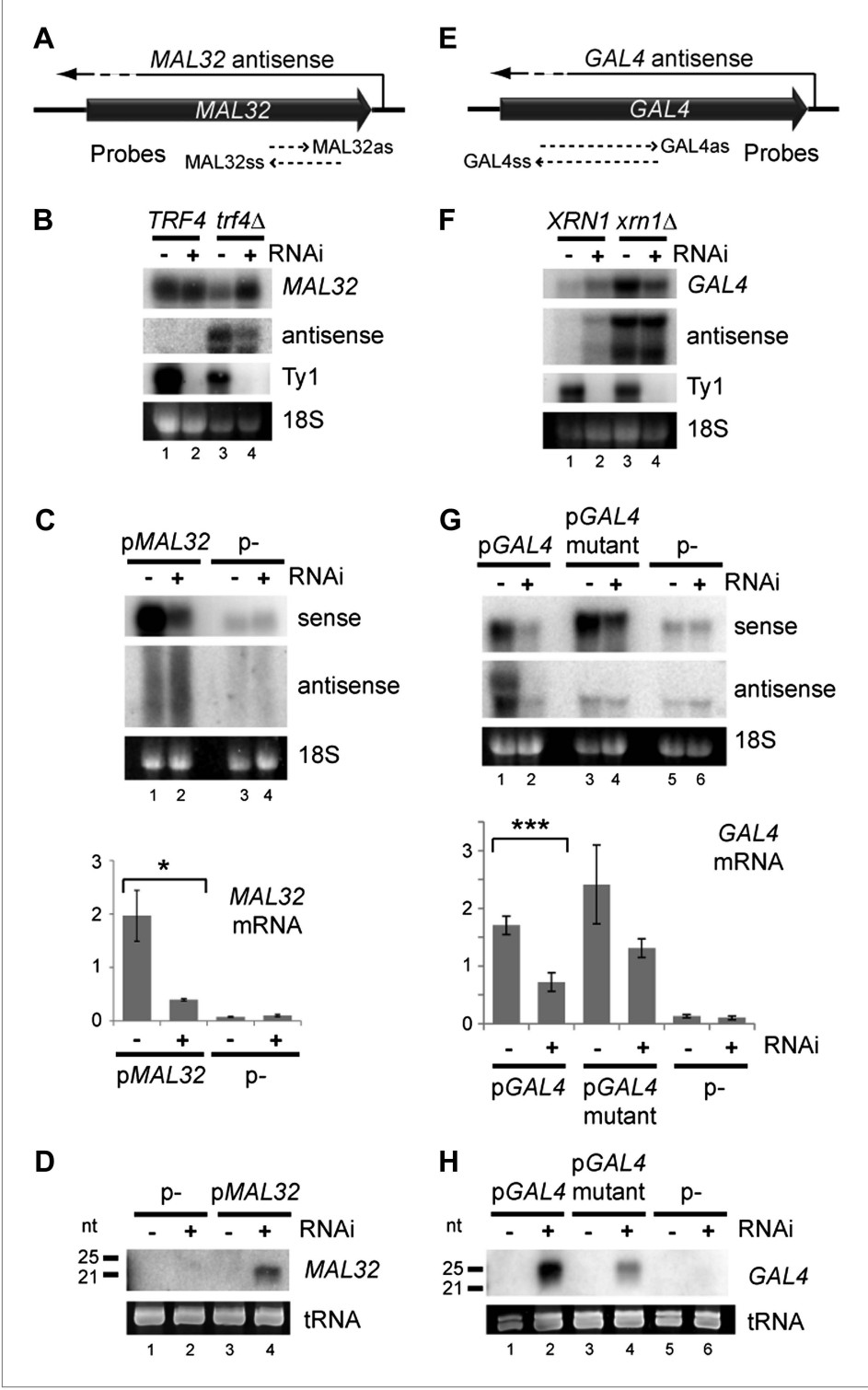

**Figure 3**. Copy number amplification of coding genes can instigate RNA interference (RNAi). (**A**) Schematic of *MAL32* locus. (**B**) *MAL32* mRNA and antisense non-protein coding RNA (ncRNA) comparing wild-type and RNAi+ strains in wild-type and *trf4Δ* backgrounds; cells grown on YP raffinose (extended image and quantification shown in *Figure 3—figure supplement 2A*). (**C**) mRNA and antisense ncRNA from *MAL32* cloned onto a high-copy plasmid in wild-type and RNAi+ strains. Lanes 3,4 show empty vector control. Antisense panel shows degradation products, no full-length antisense is detectable due to Trf4 activity. (**D**) Short interfering RNA (siRNA) analysis of *Figure 3. Continued on next page*

*Figure 3. Continued*

cells from **C**. (**E**) Schematic of *GAL4* locus. (**F**) *GAL4* mRNA and antisense ncRNA in wild-type and RNAi+ strains; cells grown in YP galactose (extended image and quantification shown in *Figure 3—figure supplement 2B*). (**G**) mRNA and antisense ncRNA from *GAL4* locus cloned onto a high-copy plasmid in wild-type and RNAi + strains. Lanes 5,6 show empty vector, signal is from genomic *GAL4*; note that cells used here are diploids to mitigate defects in galactose response (see 'Materials and methods'). Lanes 3,4 show a previously described *GAL4* antisense mutant (*Geisler et al., 2012*); this removes detectable antisense RNA for genomic *GAL4*, but the mutant sequence still expresses an antisense ncRNA when cloned on the high-copy plasmid (see *Figure 3—figure supplement 4*). (**H**) siRNA analysis of cells in **G**. For quantification, n = 4 biological replicates, error bars represent ± 1se, *p<0.05, ***p<0.01 by Student's *t* test, y axes in arbitrary units.

The following figure supplements are available for figure 3:

**Figure supplement 1**. Characterization of the *MAL32* sense/antisense system.

**Figure supplement 2**. Images and quantification of RNA interference (RNAi) degradation patterns.

**Figure supplement 3**. Verification of RNA interference (RNAi) knockdown.

**Figure supplement 4**. Expression of *GAL4* antisense RNA in mutant construct.

**Figure supplement 5**. Plasmid copy number in high-copy plasmid strains.

**Figure supplement 6**. Confirmation of RNA interference (RNAi) action by immunoprecipitation and qRT-PCR.

should be observed between total RNA abundance and siRNA abundance, and there should be no difference between distributions of single-copy loci and multi-copy loci. We observed little evidence for such a positive correlation (*Figure 4—figure supplement 1*); however, plots of siRNA versus total RNA abundance for single-copy and multi-copy loci showed strikingly different distributions, with multi-copy loci clearly biased towards higher siRNA production (*Figure 4A*). To quantify this difference, loci were segregated into eight groups of increasing total RNA abundance and siRNA abundance was assessed for single-copy and multi-copy loci in each group (*Figure 4B*). siRNA production was significantly higher from multi-copy loci than from single-copy loci in all except the lowest category of RNA abundance. This result was robust to changes in the threshold between low and high copy, and was still observed in a comparison of low to medium copy number, showing that high-copy Ty1 retrotransposons were not distorting the analysis (*Figure 4—figure supplement 2*). A normalization step is required in these analyses to deal with mapping of sequence reads to multi-copy loci (discussed in detail in 'Materials and methods'); however, the same differences were observed with no normalization or a different normalization scheme (*Figure 4—figure supplement 3*). These surprising results show that multi-copy loci produce more siRNA than single-copy loci with equivalent RNA abundance. If this observation is real, the siRNA:total RNA ratio should be predictive of copy number, an important test since this comparison requires no copy number normalization. As predicted, the 1% of genome with the highest siRNA:total RNA ratio is massively enriched for multi-copy loci (*Figure 4C*), and when this ratio was plotted across a chromosome, an obvious correlation was observed between regions of high-copy number and regions with high siRNA:total RNA abundance (*Figure 4D*). These analyses clearly demonstrate that selectivity towards the products of multi-copy loci is an emergent property of a minimal RNAi system.

We then directly tested the effect of copy number at the *MAL32* locus. We constructed strains in which *MAL32* sense and antisense RNAs were expressed at similar levels from multi-copy or single-copy loci by over-expressing single-copy sense and antisense (*Figure 5A*). In this system, both sense and antisense RNAs were produced at higher levels from the single-copy system (*Figure 5B* compare lanes 1 and 3) but more siRNAs were produced from the multi-copy system (*Figure 5C* compare lanes 2 and 4). The over-expression of both RNAs from the single-copy *MAL32* locus led to the production of easily detectable siRNA, as would be expected; however, this result directly demonstrates that gene copy number influences the formation of siRNA above and beyond the effect of total RNA abundance. The increased siRNA production in these cells is most likely due to enhanced dsRNA formation in the multi-copy system. To confirm this, we quantified *MAL32* RNA in wild-type cells that is resistant to

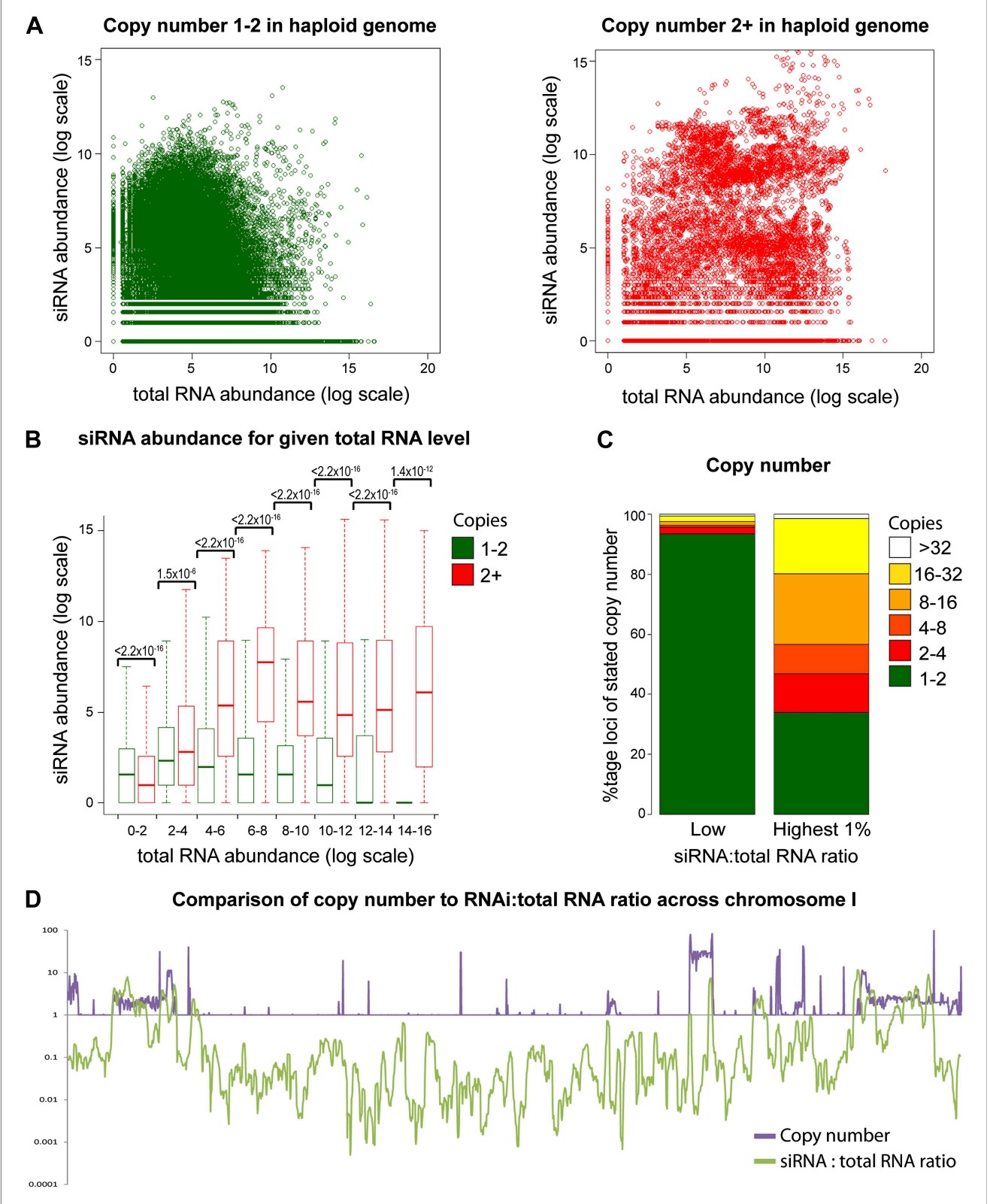

**Figure 4**. Multi-copy loci are preferentially targeted by RNA interference (RNAi). (**A**) Short interfering RNA (siRNA) (**Drinnenberg et al., 2011**) and total RNA (**Hobson et al., 2012**) abundance for loci with copy number <2 (left, single-copy) or ≥2 (right, multi-copy). (**B**) Quantification of data from **A** binned into categories of increasing total RNA level, with p values for pairwise comparisons of siRNA abundance in single-copy and multi-copy datasets using

*Figure 4. Continued on next page*

*Figure 4. Continued*

the Wilcoxon Rank Sum test. (**C**) Copy number distribution of the 1% of loci with the highest siRNA:total RNA ratio compared with other loci; difference is significant by Wilcoxon Rank Sum test, p<2.2 × 10⁻¹⁶, loci scoring below noise threshold (0–2 category in **B**) were removed. n values for tests in **B** and **C** are given in *Table 2*. (**D**) Comparison of copy number with siRNA:total RNA ratio across chromosome I.

The following figure supplements are available for figure 4:

**Figure supplement 1**. Short interfering RNA (siRNA) versus total RNA abundance.

**Figure supplement 2**. Increased short interfering RNA (siRNA) production from multi-copy sequences is robust to different cut-offs.

**Figure supplement 3**. Increased short interfering RNA (siRNA) production from multi-copy sequences is not due to copy number normalization.

the single-strand specific nuclease RNase A, and observed significantly more RNase-resistant material in cells expressing *MAL32* from the multi-copy system than the single-copy system (*Figure 5D*). This experiment shows that a multi-copy locus produces more dsRNA than an equivalently expressed single-copy locus in wild-type cells without the RNAi system, explaining the increased siRNA formation in RNAi+ cells.

The influence of copy number suggested that dsRNA formation and potentially siRNA production may occur in the nucleus. We initially tested this by immunofluorescence against Dicer and dsRNA in mixed populations of wild-type and RNAi+ cells (*Figure 6—figure supplement 1*). Dicer was present in cytoplasmic foci, but also showed a diffuse cytoplasmic staining (compare the indicated wild-type and RNAi+ cells) and, under wide-field imaging, appeared to be present in the nucleus. However, super-resolution images of the same cells showed nuclear exclusion of Dicer; therefore, if Dicer is present in the nucleus, it is at low levels. dsRNA staining in these cells revealed many cytoplasmic foci, presumably Killer virus dsRNAs that are known to be incompletely cleared by RNAi (*Drinnenberg et al., 2011*), but did not show unambiguous nuclear staining.

As an alternative, we asked whether the spatial configuration of gene copies within the nucleus could affect siRNA formation; such an effect would provide strong evidence for the formation of dsRNA in the nucleus. In systems which undergo efficient RNAi such as the rDNA and 2μ plasmids, all gene copies are clustered together in a small sub-nuclear volume. To test the importance of this clustering, we used the sense-antisense system at the *TRP1* locus which produces detectable siRNA even when present in the genome in only two tandem copies (data not shown). We generated strains with three tandem copies of *TRP1* on a single plasmid (Clustered, Cls) or three unlinked copies (Dispersed, Dsp) (*Figure 6A*), and expressed Dicer without Argonaute to allow siRNA formation but minimize the effect of RNAi on total RNA levels. Quantification of total RNA showed that both systems produced similar amounts of sense and antisense RNA molecules (*Figure 6B*), although this experiment was complicated by read-through transcripts of antisense *TRP1* from the clustered system (*Figure 6B* lanes 1,2), a behavior that was not prevented by insertion of transcriptional stop cassettes between the repeats. Nonetheless, the clustered system produced fourfold more *TRP1* siRNA than the dispersed system (*Figure 6C*), showing that close nuclear juxtaposition of transcriptional loci enhances dsRNA formation.

While testing the effects of copy number amplification on siRNA production, we noticed that even low abundance sense-antisense ncRNA pairs (selected from a published dataset, *Xu et al., 2009*) underwent efficient RNAi when amplified to high copy number. For both the SUT176 and SUT430 systems (*Figure 7A*), the sense and antisense RNAs are barely detectable by northern blot and are clearly not targeted for degradation by RNAi (*Figure 7B*). However, after cloning on high-copy plasmids, the full-length RNAs became highly susceptible to RNAi and produced copious siRNAs (*Figure 7C,D*). This raised the interesting prospect that low abundance pervasive transcription would be sufficient to trigger efficient RNAi responses from sequences that undergo copy number amplification.

Clear examples of pervasive transcription are not well defined in any organism because, by definition, the products of pervasive transcription are almost undetectable. We therefore chose to examine the GAL gene cluster (*Figure 8A*), which is tightly transcriptionally repressed in cells grown in glucose. Under these conditions, antisense ncRNAs are produced from the *GAL10* ORF with a known steady-state abundance of one RNA molecule per 14 cells (*Houseley et al., 2008*; *Pinskaya et al., 2009*) (arrows in *Figure 8B* lane 1). Transcription of these ncRNAs is abrogated in a previously described

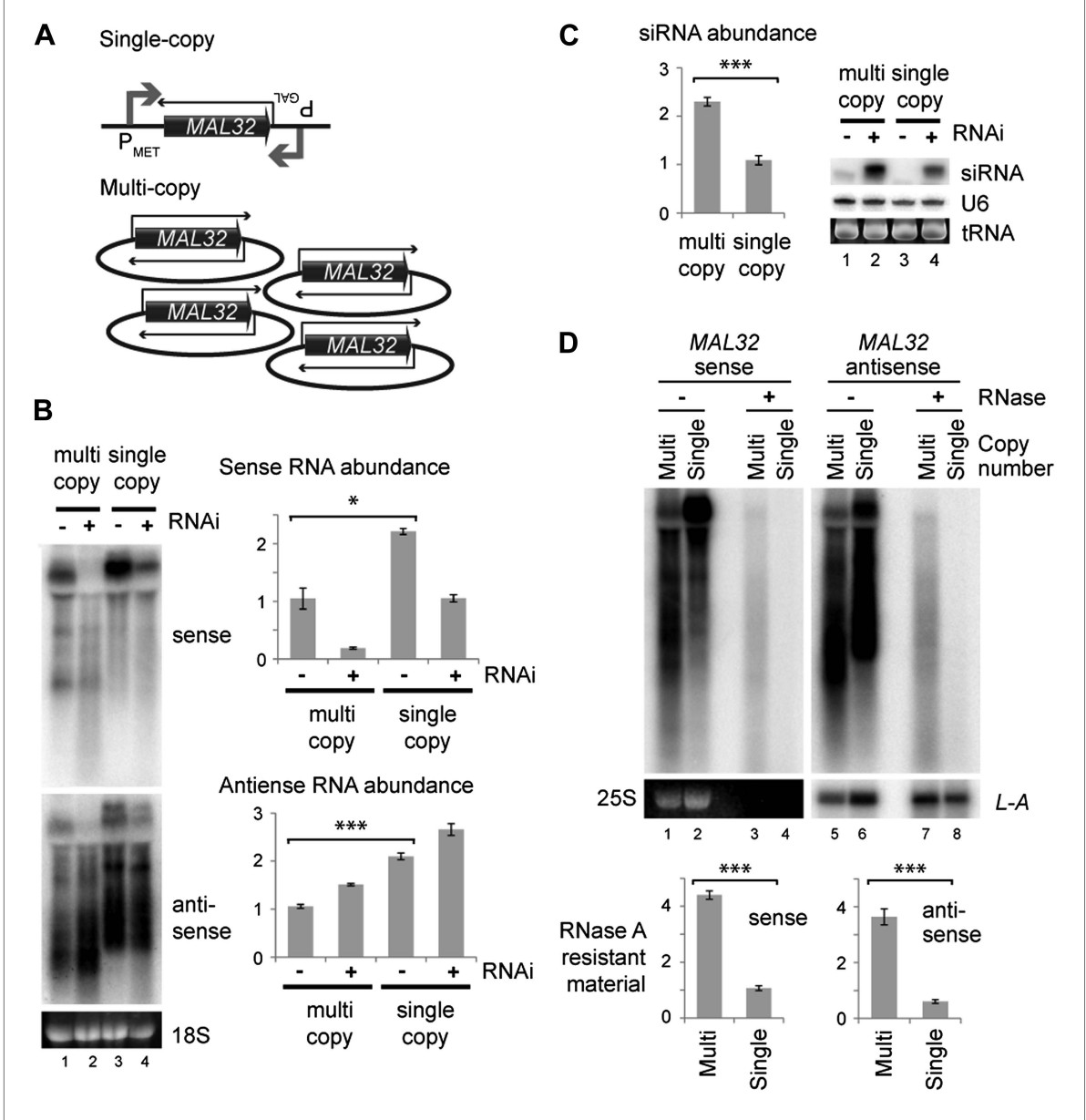

**Figure 5**. Single gene analysis of copy number effect on RNA interference (RNAi). (**A**) Schematic of single-copy and multi-copy *MAL32* system. (**B**) Northern analysis of *MAL32* RNA from single-copy and multi-copy systems. All visible species are included in antisense quantification. (**C**) *MAL32* short interfering RNA (siRNA) abundance from cells in **B**. (**D**) RNase A sensitivity of *MAL32* in wild-type cells expressing multi-copy and single-copy *MAL32*. Cells were lysed on ice, treated with RNase A as indicated and analyzed by northern blot. 25S and *L-A* (a double stranded RNA) are shown as controls for loading and RNase specificity. n = 3 biological replicates, error bars ±1se, *p<0.05, ***p<0.01 by Student's *t* test, y axes in arbitrary units.

Reb1 binding site mutant (RBSΔ), leaving almost no detectable RNAs from this locus (*Figure 8B* lane 3). For reasons that remain unclear, the GAL cluster is slightly de-repressed in the RNAi+ strain (*Figure 8B* compare lanes 1,2); nonetheless, the RBSΔ RNAi+ strain (*Figure 8B* lane 4) only produces very low level heterogeneous transcripts from *GAL10*, suggesting that it forms a good model of pervasive transcription. Cloning either wild-type or RBSΔ GAL clusters onto high-copy plasmids substantially increased the levels of detectable ncRNA as expected (*Figure 8B* lanes 5,7), and these ncRNAs were processed into easily detectable siRNAs (*Figure 8C* lanes 6,8). Therefore, ncRNAs produced at the level of pervasive transcription are sufficient to mediate extensive siRNA production when the copy number of the transcribing locus is increased.

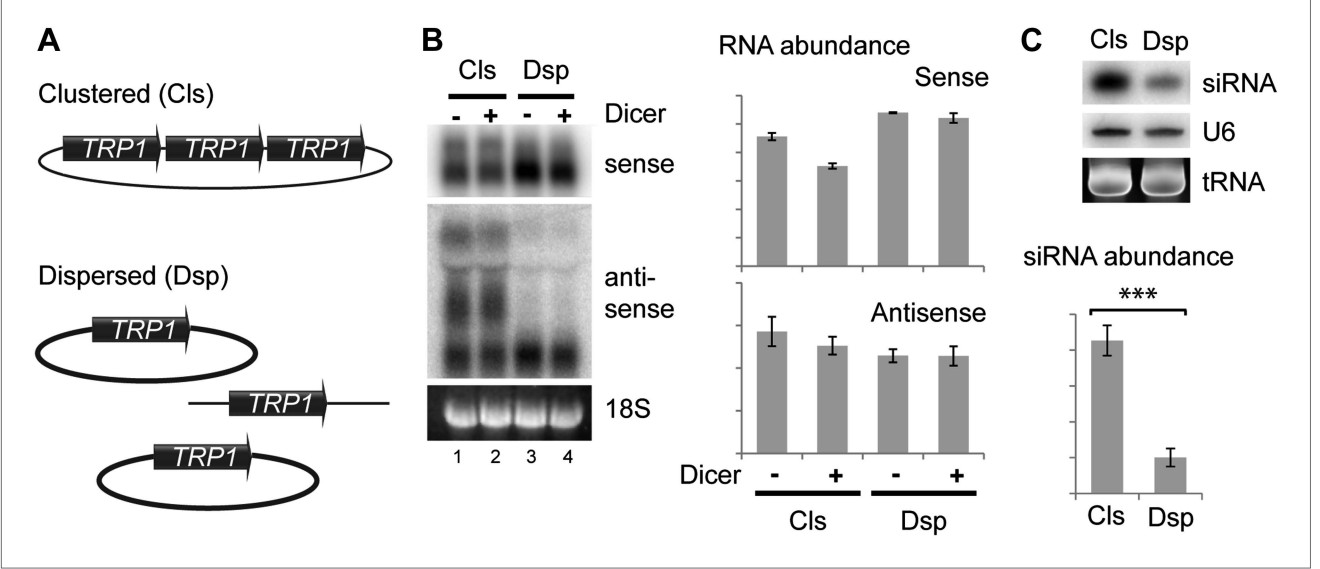

**Figure 6**. Clustered loci show higher efficiency of short interfering RNA (siRNA) formation. (**A**) Comparison of the systems used. Three copies of *TRP1* were placed either in tandem on a single-copy plasmid (Clustered, Cls) or a single copy was left in the genome at the *TRP1* locus and two further copies were placed on different single-copy plasmids (Dispersed, Dsp). Dicer was expressed from a single-copy plasmid. (**B**) Sense and antisense RNA expression in clustered and dispersed systems with and without Dicer. Quantification shows that Dicer alone has little effect on total RNA levels. Read-through species visible in lanes 1,2 are included in the quantification; values have been normalized for the different number of probe binding sites in the read-through RNAs. In the absence of this normalization (i.e., counting the number of binding sites rather than the number of molecules), the clustered antisense is approximately twofold more abundant than the dispersed antisense, which is insufficient to explain the difference in siRNA formation. (**C**) siRNA produced from *TRP1* in clustered and dispersed systems. n = 3 biological replicates, error bars ±1se, ***p<0.01 by Student's *t* test, y axes in arbitrary units.

The following figure supplements are available for figure 6:

**Figure supplement 1**. Analysis of Dicer and double stranded RNA (dsRNA) localization.

The siRNAs produced from the high-copy *GAL10* locus are clearly sufficient to degrade the *GAL10* ncRNAs in the RNAi+ background (***Figure 8B*** compare lanes 5,7 with lanes 6,8); however, a classical RNAi response should be able to degrade RNA expressed from a separate locus. To test this we introduced the high-copy GAL cluster plasmids into a strain in which the single-copy genomic *GAL10* ORF is expressed at high levels from a $Cu^{2+}$-dependent promoter, allowing expression of the *GAL10* mRNA from the single-copy locus while the GAL clusters present on the high-copy plasmids remain fully repressed. As observed for the *GAL10* ncRNAs, the *GAL10* mRNA was expressed at higher levels in the RNAi+ strain than in the wild-type (***Figure 8D*** compare lanes 1,2) but, nonetheless, both wild-type and RBSΔ high-copy GAL cluster plasmids caused highly significant >50% knockdowns of the *GAL10* mRNA compared with the empty vector control (***Figure 8D*** lanes 2,4,6). This was not an indirect effect of the high-copy GAL clusters alone as, in the wild-type background, *GAL10* mRNA levels were slightly increased by the presence of the GAL plasmids (***Figure 8D*** lanes 1,3,5). These data demonstrate that pervasive transcription of a high-copy locus is sufficient to instigate an effective RNAi response that can mediate the degradation of a target mRNA in trans.

## Discussion

The ability of the RNAi system to selectively target the products of high-copy sequences such as transposons provides a remarkably efficient genome defense, as well as an effective way to differentiate heterochromatic regions, which are often repetitive, from gene-rich euchromatin. Here we have demonstrated that RNAi has an innate preference for the products of high-copy sequences, probably because RNA from high-copy sequences forms dsRNA more efficiently. It has long been known that cells can recognize and silence high-copy DNA, which would form a basic defense against uncontrolled amplification of transposable elements (reviewed in ***Hsieh and Fire, 2000***). This of course requires a mechanism to count copy number, or at least differentiate high- and low-copy regions, which has

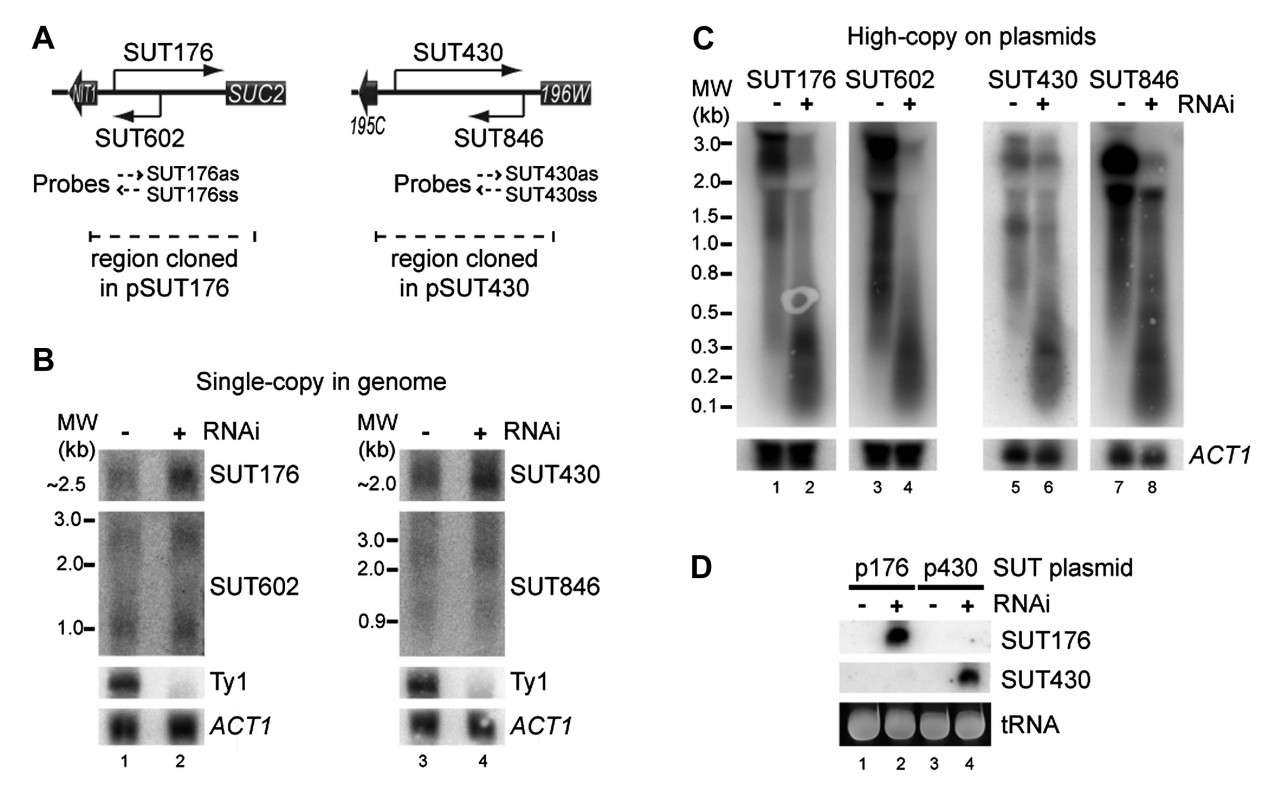

**Figure 7**. RNA interference (RNAi) against transcripts from amplified low-expression systems. (**A**) Schematic diagrams of SUT176 and SUT430 loci. (**B**) Northern analysis of SUT176 and SUT430 transcripts from single-copy genomic loci in wild-type and RNAi+ cells. Ty1 RNA is a positive control for RNAi, *ACT1* is a loading control. (**C**) Analysis of SUT176 and SUT430 non-protein coding RNAs (ncRNAs) expressed from high-copy plasmids in wild-type and RNAi+ cells. Amplified regions are indicated in **A**. (**D**) Short interfering RNA analysis of cells in **C**.

remained mysterious. Our data show that RNAi provides such a mechanism by selectively targeting the products of high-copy loci.

The production of siRNA from high-copy DNA, which would be the basis of such a counting mechanism, absolutely requires that all DNA is transcribed; if this does not occur, transposable elements that remain transcriptionally silent would be invisible to the system. Pervasive transcription, the general background of very low level RNA produced across the genome, ensures that the vast majority of the genome is transcribed, and therefore that no region remains completely silent (*Cheng et al., 2005*; *Birney et al., 2007*; *Kapranov et al., 2007*; *Goodman et al., 2012*). The extent to which mammalian genomes are pervasively transcribed has been controversial; however, many of the questions revolve around whether the pervasive transcripts represent defined functional products or whether much of the detected RNA represents random transcriptional noise (*van Bakel et al., 2010*; *Clark et al., 2011*). For a general surveillance function, it does not really matter whether pervasive transcription is formed of many discrete transcripts or occurs at random since either process should be sufficient to generate dsRNA. If a large proportion of pervasive transcription does represent random noise, this would be actively advantageous; random transcription would be sequence independent, and therefore transposable elements could not become fully silent by mutating individual promoter sequences. We suggest that the primary function of pervasive transcription lies in ensuring the whole genome is transcribed to allow identification and suppression of transposable elements; this does not conflict with the idea that some proportion of these transcripts may have additional functions.

We propose that hybridization kinetics explains the dependence of RNAi on copy number (shown in *Figure 9*). The rate of formation of dsRNA from single stranded sense and antisense RNA is proportional to the concentration of each strand of RNA, and so is inversely proportional to the square of the reaction volume. Technically, the reaction volume is the non-excluded volume of the cell; however, this assumes

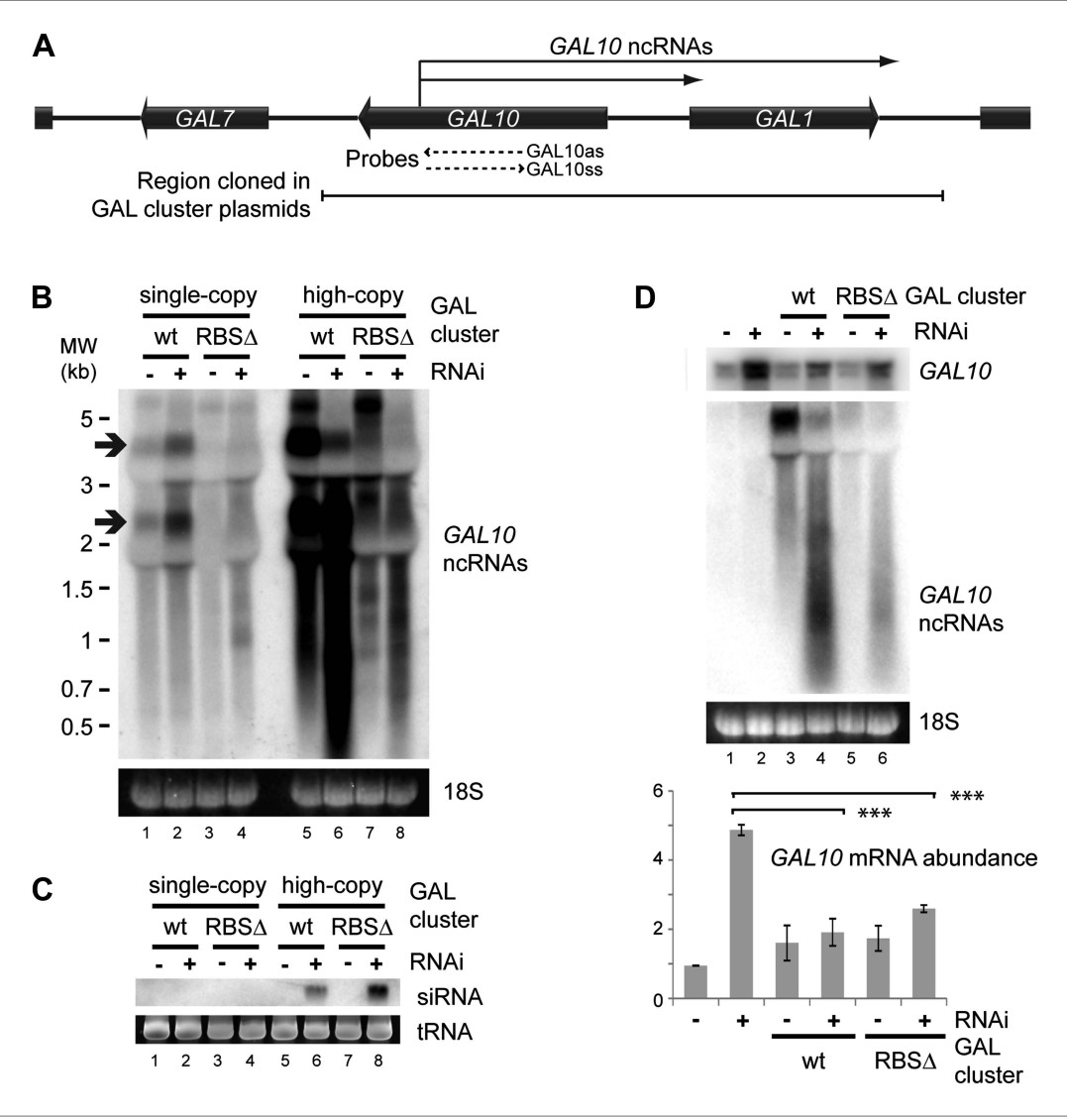

**Figure 8**. RNA interference (RNAi) against pervasive transcripts from the repressed GAL cluster. (**A**) Schematic representation of the GAL cluster. (**B**) Non-strand-specific northern blot of non-protein coding RNAs (ncRNAs) produced from the GAL locus present at single-copy (lanes 1–4) or high-copy (lanes 5–8), showing wild-type and RBSΔ mutant. Arrow indicates *GAL10* antisense RNA. Strand-specific northern blots for the same RNA are shown in *Figure 8—figure supplement 1*. (**C**) *GAL10* short interfering RNA (siRNA) from the same cells as in **B**. (**D**) Expression of *GAL10* mRNA from a single-copy genomic locus under the control of a $Cu^{2+}$-responsive promoter in wild-type and RNAi+ strains carrying an empty vector (lanes 1,2), high-copy wild-type GAL cluster (lanes 3,4), or high-copy RBSΔ GAL cluster (lanes 5,6). n = 3 biological replicates, error bars ±1se, ***p<0.01 by Student's *t* test, y axes in arbitrary units.

The following figure supplements are available for figure 8:

**Figure supplement 1**. Strand-specific analysis of *GAL10* non-protein coding RNAs (ncRNAs).

---

a uniform distribution of RNA throughout the cell. In reality the RNA is far from evenly distributed, so some small volumes may have very high concentrations of RNA and, within these volumes, the rate of dsRNA formation will be dramatically higher than in the bulk of the cell. Single-copy loci cannot simultaneously transcribe sense and antisense RNA (*Hobson et al., 2012*) so, although such a locus can give rise to a mixed population of sense and antisense RNA in the cytoplasm over time, in the vicinity of the transcription site only one sense of RNA should ever be present, assuming efficient RNA export. Annealing of sense and antisense RNA must therefore occur in the relatively large

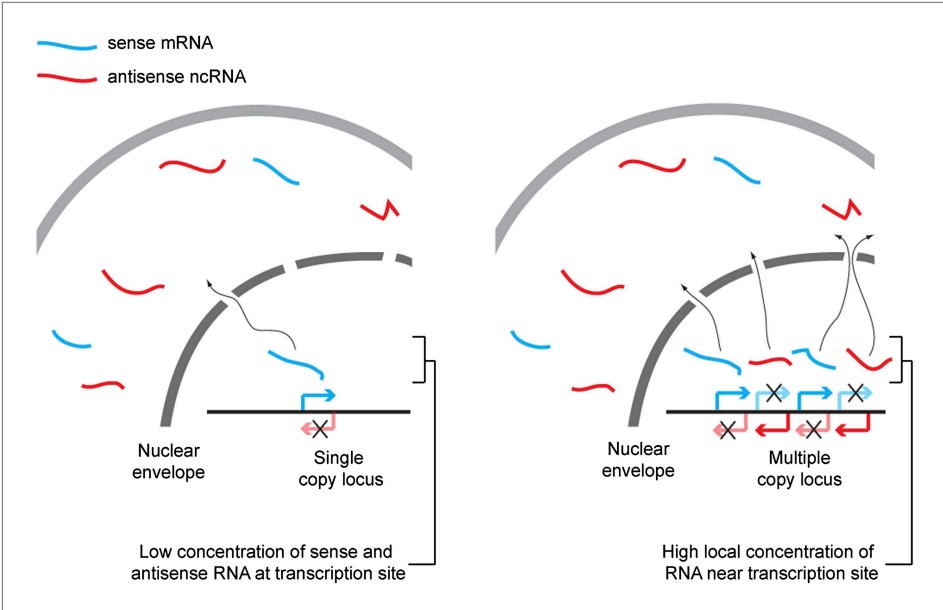

**Figure 9**. Proposed mechanism for RNA interference (RNAi) on high-copy loci. The rate of double stranded RNA (dsRNA) formation, the required first step in RNAi, is highly dependent on local RNA concentration. Single-copy loci cannot simultaneously transcribe sense and antisense RNA, allowing RNA export to occur before the strands meet and requiring hybridization to occur in the relatively large cytoplasmic volume. In contrast, sense and antisense RNA can be simultaneously transcribed from different parts of a multi-copy locus and, therefore, if the copies are closely juxtaposed in the genome or in 3D space, the local concentration of sense and antisense RNA around the transcription sites should be high, favoring dsRNA formation.

non-excluded cytoplasmic volume of the cell, which will be inefficient except for very highly expressed RNA. In contrast, at multi-copy loci the simultaneous production of sense and antisense RNA from many closely spaced sites can lead to high concentrations of sense and antisense RNA around the transcription site, causing efficient duplex formation and hence efficient RNAi.

This mechanism predicts that dsRNA formation should occur in the nucleus, but we were not able to detect Dicer in the nucleus by immunofluorescence. This reflects the situation in higher eukaryotes where Dicer is largely cytoplasmic, but recent experiments in Drosophila and mammalian cells have detected small quantities of nuclear Dicer, particularly associated with chromatin (*Sinkkonen et al., 2010*; *Cernilogar et al., 2011*; *Gullerova et al., 2011*; *Doyle et al., 2013*). Low but functional levels of Dicer may therefore be present in the nucleus of RNAi+ cells and able to generate siRNA. Alternatively, the dsRNA may be exported and processed in the cytoplasm. To our knowledge, there is no clear evidence for or against export of dsRNA by normal pathways; certainly these would not be too large or too structured to pass through nuclear pores compared, for example, with pre-ribosomes.

One notable prediction of this mechanism is that clustering of multi-copy transcription sites would be a particularly efficient way to increase the local density of sense and antisense RNA. All the systems we have described in this paper are clustered: rDNA repeats are arranged in tandem, telomeres are known to cluster together at various points in the cell cycle (*Gotta et al., 1996*), and high-copy 2μ plasmids exist in a discrete focus that is vital for copy number maintenance (*Velmurugan et al., 2000*; *Wong et al., 2002*). The comparison of siRNA formation from clustered and dispersed *TRP1* loci provides experimental evidence for this effect since, for a given quantity of sense and antisense RNA, the clustered system produces more siRNA. Although the clustered system also produces read-through transcripts, these would not have a higher hybridization rate than the non-read-through RNAs as the hybridization rate depends on the frequency of collisions between molecules. Intriguingly, Tf2 retrotransposons in *Schizosaccharomyces pombe* are clustered by the action of centromere protein B (CENP-B), which also silences these elements through histone deacetylation (*Cam et al., 2008*). This clustering would allow cells to produce siRNA against the Tf2 elements through pervasive transcription, although multiple mechanisms silence Tf2 retro-transposons (*Yamanaka et al., 2012*), providing an extra defense against retrotransposon activation.

Similarly, gypsy retrotransposons in Drosophila are known to cluster (*Gerasimova et al., 2000*), which may again facilitate siRNA production. Hence, the clustering of transposable elements by factors such as CENP-B would facilitate their recognition by RNAi and allow for selective RNA degradation.

Mammalian germline cells are replete with small RNAs including endogenous siRNA (*Watanabe et al., 2006*, *2008*; *Tam et al., 2008*; *Song et al., 2011*), and siRNAs in sperm and oocytes show a pronounced bias towards high-copy sequences that would be effectively explained by the selectivity of the RNAi system towards the products of high-copy loci (*Watanabe et al., 2006*; *Song et al., 2011*). However, it remains unclear how some dsRNA precursors of siRNAs are generated, particularly for retrotransposons that are primarily expressed only on the sense strand. We suggest that pervasive transcription would provide sufficient antisense RNA for this role, just as we observed for high-copy GAL cluster sequences in yeast.

In comparison to the germline, the response of mammalian somatic cells to dsRNA is distinctly muted. dsRNA could be processed into siRNA, be altered by RNA editing (*Hogg et al., 2011*), or could activate the interferon response leading to apoptosis (*Gantier and Williams, 2007*). However, transgenic mice expressing a hairpin dsRNA construct produce minimal siRNAs, little edited RNA, and show no phenotype that might indicate cell death (*Nejepinska et al., 2012a*). Nonetheless, siRNAs produced from LINE-1 retrotransposons have been detected in cell culture (*Yang and Kazazian, 2006*), and high-copy transfected plasmids expressing a sense-antisense RNA pair do produce detectable siRNA in HEK293 cells (*Nejepinska et al., 2012b*). This shows that a basic siRNA response with an apparent bias towards high-copy sequences is functional in mammalian somatic cells.

## Materials and methods

### Yeast strains, plasmids and culture conditions

Yeast deletion strains (*Supplementary file 1*) were created by standard methods using the oligonucleotides in *Supplementary file 1*. Plasmids are described in *Supplementary file 1* with construction details. Cells were grown on rich media (2% peptone, 1% yeast extract, 2% sugar) or synthetic media (0.69% yeast nitrogen base with ammonium sulfate, amino acids, 2% sugar) for plasmid assays. *GAL10* mRNA was induced with 20 µM $CuSO_4$ in *Figure 8D*. Media components were purchased from Formedium. Cells were grown to mid-log (OD 0.4–0.6) at 30°C for most experiments or at 25°C for experiments involving *trf4Δ* mutants. The W303 background strain used here has defects in galactose induction in synthetic media, so strains in *Figure 3G,H* were diploids of W303xBY4741 that show a normal galactose response.

### RNA extraction and northern analysis

RNA was extracted by three procedures described below. High molecular weight RNA was prepared using the hot phenol method for all experiments except *Figures 7B, 3B, 3F* and *Figure 3—figure supplements 1,2* where guanidine thiocyanate (GTC)-phenol preparations were used. 5–10 µg glyoxylated RNA was resolved on 1.2% gels as described (*Sambrook and Russell, 2001*), transferred to Hybond N+ membrane (GE) and hybridized with probes listed in *Supplementary file 1* using either Church Hyb (*Sambrook and Russell, 2001*) or UltraHyb (Life Technologies). RNA probes were hybridized at 65°C and washed at 65°C using 0.1× SSC, 0.1% SDS, DNA probes in Church Hyb were hybridized at 65°C and washed at 65°C with 0.5× SSC, 0.1% SDS, DNA probes in UltraHyb were hybridized at 42°C and washed at 55°C using 0.2× SSC, 0.1% SDS. Small RNA enriched fractions were isolated using the mirVANA kit (Ambion). 4–10 µg small RNA was separated on 15% polyacrylamide/8 M urea gels containing 20 mM MOPS or 1× TBE, transferred in 20 mM MOPS or 0.5× TBE to Hybond N membrane (GE) and chemically cross-linked as described (*Pall and Hamilton, 2008*). We observed no difference in cross-linking efficiency between MOPS and TBE gels, but resolution of TBE gels was superior in our hands. siRNAs were detected using random primed probes (*Supplementary file 1*) hybridized in UltraHyb Oligo (Life Technologies) at 42°C and washed with 2× SSC, 0.5% SDS at 42°C, U6 control oligonucleotide was labeled using T4 polynucleotide kinase and hybridized in Church Hyb under the same conditions.

### Hot phenol RNA preparation

$10 \times 10^7$ cells in 15 ml tubes were re-suspended in 600 µl TES (10 mM Tris pH 7.5, 10 mM EDTA, 0.5% SDS) and 600 µl phenol pH 7. The mixture was incubated at 65°C for 20 min with 30 s vortexing every 5 min, before briefly chilling on ice. Samples were centrifuged for 5 min and the upper phase extracted. This phase (5–600 µl) was extracted twice with phenol:chloroform (5:1) and once with chloroform

before precipitation with 50 µl 3 M sodium acetate (NaOAc) pH 5.2 and 1.1 ml ethanol. The pellet was washed with 70% ethanol and re-suspended in 30 µl water.

## GTC-phenol RNA preparation

$2 \times 10^7$ cells were lysed by 5 min vortexing at 4°C with 50 µl glass beads and 40 µl GTC-phenol (2.1 M GTC, 26.5 mM Na citrate pH7, 5.3 mM EDTA, 76 mM β-mercaptoethanol, 1.06% N-lauryl sarcosine, 50% phenol pH7). 600 µl GTC-phenol was added, mixed, and samples were heated at 65°C for 10 min, then placed on ice for 10 min. 160 µl 100 mM NaOAc pH 5.2 and 300 µl chloroform:isoamyl alcohol (24:1) were added, samples were vortexed and centrifuged at top speed for 5 min. The upper phase was extracted, precipitated with 1 ml ethanol, washed with 70% ethanol and the pellet re-suspended in 6 µl water. 3 µl RNA was analyzed per lane.

## Small RNA purification

Small RNAs were isolated using a mirVANA kit (Life Technologies) with minor modifications. $35 \times 10^7$ cells were thoroughly re-suspended in 100 µl lysis/binding buffer, 200 µl glass beads were added, and the samples were vortexed for 5 min at 4°C. 500 µl lysis/binding buffer were added and the samples were mixed before proceeding with the isolation as per the manufacturer's instructions, starting with addition of the miRNA homogenate additive. After isolation the samples were generally re-precipitated and re-suspended in 20 µl water.

## RNase A treatment

$20 \times 10^7$ cells were harvested and split into two aliquots, then re-suspended in 600 µl 10 mM Tris pH 7.5, 10 mM EDTA on ice. Cells were lysed with glass beads (10 cycles of 30 s vortex, 60 s on ice), and 5 µg RNase A was added to one aliquot followed by 30 min incubation on ice. After centrifugation for 10 min at 4500×$g$, 600 µl lysate was extracted, SDS added to 0.5%, and RNA extracted by the hot phenol method as above. RNase A treated samples were re-suspended in 12 µl water.

## DNA isolation and Southern blotting

Cells from 2 ml saturated culture were washed with 50 mM EDTA, then spheroplasted with 250 µl 0.34 U/ml lyticase (Sigma L4025) in 1.2 M sorbitol, 50 mM EDTA, 10 mM DTT. After centrifuging at 1000×$g$, the cells were gently resuspended in 400 µl of 0.3% SDS, 50 mM EDTA, 100 µg/ml RNase A and incubated at 37°C for 30 min. 4 µl of 20 mg/ml proteinase K was added and the samples were mixed by inversion and heated to 65°C for 30 min. 160 µl 5 M potassium acetate was added after cooling to room temperature, the samples were mixed by inversion and then chilled on ice for 30 min. After 10 min centrifuging at 10,000×$g$, the supernatant was poured into a new tube containing 500 µl phenol:chloroform pH 8 and the samples were mixed on a wheel for 15 min. The samples were centrifuged for 10 min at 10,000×$g$ and the upper phase was extracted using cut tips and precipitated with 400 µl isopropanol. Pellets were washed with 70% ethanol, air-dried and left overnight at 4°C to dissolve in 30 µl TE. After gentle mixing, 10 µl of each sample was digested with 40 U of EcoRV, ethanol precipitated and separated on a 25 cm 1% TBE gel at 90 V overnight. The gel was washed in 0.25 N HCl for 15 min, 0.5 N NaOH for 45 min, and twice in 1.5 M NaCl 0.5 M Tris pH 7.5 for 20 min before transfer to HyBond N+ membrane in 6× SSC. The membrane was probed for *URA3* and *GAL7* in Church Hyb at 65°C and washed with 0.5× SSC, 0.1% SDS at 65°C.

## Immunofluorescence

Cells were grown to OD 0.5 in YPD and the cultures mixed as required, 4 ml per coverslip. The cells were fixed with 440 µl 37% formaldehyde (Merck, microscopy grade) for 15 min at room temperature, then centrifuged for 2 min at 4600 rpm. The cells were washed three times with 1 ml of buffer B (0.1 M sodium phosphate pH 7.5, 1.2 M sorbitol), then re-suspended in 100 µl buffer B containing 3 µl 17 U/µl lyticase (Sigma L2524) and 10 mM DTT for 15 min. The cells were centrifuged for 2 min at 1000×$g$, then washed with 1 ml buffer B. The cells were re-suspended in 40 µl buffer B, applied to a poly-L-lysine coated coverslip (Zeiss 18 × 18 × 0.170 ± 0.005 mm) and left for 20 min before washing twice with buffer B. Coverslips were treated with −20°C methanol for 6 min, then dipped in −20°C acetone for 10 s, followed by two washes with PBS. Coverslips were blocked for 30 min with 5% milk 0.3% Triton-X100 in PBS, washed with PBS, then incubated overnight at 4°C with primary antibodies in 50 µl 1% BSA 0.3% Triton-X100 in PBS. Coverslips were washed three times with PBS and incubated for 30 min at room temperature with secondary antibodies 1:1000 in same buffer as primaries. After washing three times with PBS, the

samples were dehydrated with 70%, 90%, and 100% ethanol and mounted in Pro-long Gold with DAPI (Life Technologies). Antibodies were rabbit anti-GFP (Life Technologies A11122) at 1:500 and mouse anti-dsRNA J2 (ESC 10010200) at 1:1000. Images were acquired using a Nikon N-SIM microscope comprising a Nikon Ti-E microscope, Nikon 100× 1.49 NA lens, Nikon SIM illumination module, and Andor iXon 897 EM-CCD camera. SIM data were acquired in '3D-SIM' mode using five phases and three rotations. DAPI, Alexa Fluor 488, and Alexa Fluor 594 dyes were excited using 405, 488, and 561 nm laser light, respectively. Super-resolution images were reconstructed using Nikon Elements software. Equivalent wide-field images were reconstructed in FIJI (ImageJ, NIH) by summing the phase shifts from one grid rotation.

## Imaging and analysis

Gels and phosphor screens were imaged using FLA 3000 (Fuji) or FLA 7000 (GE) systems. Quantification was performed using AIDA (Fuji) or ImageQuant (GE), depending on the scanner used. Images were prepared for publication with ImageJ by smoothing and minimal contrast enhancement. Images from the FLA3000 had a Gamma Shift of 3 applied.

## 5′ RACE

5′ RACE was performed with an ExactSTART Eukaryotic mRNA 5′- and 3′-RACE Kit (Epicentre) as per manufacturer's instructions, except that reverse transcription was primed from random hexamers.

## RNA immunoprecipitation

50 ml of cells at $0.7 \times 10^7$ cells/ml were harvested, washed, and frozen on nitrogen. Cells were thawed, washed twice in 1 ml lysis buffer (50 mM HEPES pH7.5, 50 mM KCl, 5 mM MgCl$_2$, 1 mM DTT, 1× complete protease inhibitors (Roche)) and transferred to 2 ml tubes, then re-suspended in 300 µl lysis buffer. 300 µl zirconium beads were added and cells were lysed by vortexing 10 × 30 s with 30 s on ice between cycles. The lysate was clarified by centrifuging twice for 5 min at 14,000 rpm, a 12 µl aliquot was removed for total RNA and the remaining lysate was split in half and 2.5 µl mouse anti-dsRNA J2 (ESC 10010200) added to one aliquot. Antibody was bound for 2 hr at 4°C, then 20 µl GammaBind beads (GE) pre-blocked overnight with 1% BSA were added and incubation continued on a wheel for 2 hr at 4°C. The beads were washed 5× for 10 min with 1 ml wash buffer (10 mM Tris pH 7.5, 120 mM NaCl, 5 mM MgCl$_2$, 0.1% NP-40, 1 mM DTT). RNA was extracted from the beads and total lysate samples using Tri-reagent (Sigma) according to the manufacturer's instructions. 1 µg total lysate and whole immunoprecipitation samples were treated with RQ1 DNase (Promega), purified by phenol:chloroform and ethanol precipitation, then reverse transcribed from random hexamers using Superscript III (Life Technologies). Quantitative PCR was performed using Maxima SYBR qPCR mix (Fermentas).

## Bioinformatics

### Antisense analysis

Locations of XUTs (*van Dijk et al., 2011*) were merged with CUT and SUT locations (*Xu et al., 2009*), along with expression validated ORFs (*Xu et al., 2009*), and overlap between ORFs and other features calculated using an R script.

### siRNA analysis

Sequencing data for mRNA and siRNA fractions in RNAi strains (GEO accession GSE31300) (*Drinnenberg et al., 2011*) and W303 total RNA (GEO accession GSE38383) (*Hobson et al., 2012*) were quality and adapter trimmed using Trim Galore (v0.2.3; default options; http://www.bioinformatics.babraham.ac.uk/projects/trim_galore/) and aligned to the yeast genome (build SGD1.01) using Bowtie (*Langmead et al., 2009*) (v0.12.7; default settings plus '--best') allowing non-unique sequences to be assigned at random. For expressed gene analysis (*Figure 1*), reads overlapping each ORF were binned and only ORFs with >100 reads were used. For siRNA profiles (*Figure 2—figure supplement 1*), reads were binned over 50 bp intervals using SeqMonk (http://www.bioinformatics.babraham.ac.uk/projects/seqmonk/). For siRNA versus total RNA expression (*Figure 4*), read counts were quantified in consecutive 100 bp bins across the genome using SeqMonk, bins with >10,0000 total RNA reads were excluded as were bins derived from 2µ sequence which is single copy in the genome sequence but high copy in reality, and a pseudocount of one read was added to total and siRNA read counts for each bin. Total RNA levels were multiplied by copy number to correct for the division of reads amongst copies that occurs during mapping, or alternate normalization was applied in *Figure 4—figure supplement 2* (see below

**Table 2.** n values for statistical tests

| Figure | Category | n | Category | n |
|---|---|---|---|---|
| 4B | 0–2 single-copy | 19,892 | 0–2 multi-copy | 1128 |
| | 2–4 single-copy | 36,474 | 2–4 multi-copy | 1111 |
| | 4–6 single-copy | 34,921 | 4–6 multi-copy | 1177 |
| | 6–8 single-copy | 16,208 | 6–8 multi-copy | 1356 |
| | 8–10 single-copy | 4060 | 8–10 multi-copy | 1424 |
| | 10–12 single-copy | 922 | 10–12 multi-copy | 1505 |
| | 12–14 single-copy | 270 | 12–14 multi-copy | 887 |
| | 14–16 single-copy | 56 | 14–16 multi-copy | 199 |
| 4C | Bulk 1–2 copies | 85,495 | Top 1% 1–2 copies | 252 |
| | Bulk 2–4 copies | 1419 | Top 1% 2–4 copies | 95 |
| | Bulk 4–8 copies | 544 | Top 1% 4–8 copies | 98 |
| | Bulk 8–16 copies | 981 | Top 1% 8–16 copies | 238 |
| | Bulk 16–32 copies | 1978 | Top 1% 16–32 copies | 212 |
| | Bulk >32 copies | 695 | Top 1% >32 copies | 25 |
| 4—Supplement 1A | 0–2 single-copy | 19,892 | 0–2 multi-copy | 836 |
| | 2–4 single-copy | 36,474 | 2–4 multi-copy | 906 |
| | 4–6 single-copy | 34,921 | 4–6 multi-copy | 724 |
| | 6–8 single-copy | 16,208 | 6–8 multi-copy | 387 |
| | 8–10 single-copy | 4060 | 8–10 multi-copy | 189 |
| | 10–12 single-copy | 922 | 10–12 multi-copy | 167 |
| | 12–14 single-copy | 270 | 12–14 multi-copy | 128 |
| | 14–16 single-copy | 56 | 14–16 multi-copy | 29 |
| 4—Supplement 1B | 0–2 low-copy | 20,728 | 0–2 high-copy | 292 |
| | 2–4 low-copy | 37,380 | 2–4 high-copy | 205 |
| | 4–6 low-copy | 35,645 | 4–6 high-copy | 453 |
| | 6–8 low-copy | 16,595 | 6–8 high-copy | 969 |
| | 8–10 low-copy | 4249 | 8–10 high-copy | 1235 |
| | 10–12 low-copy | 1089 | 10–12 high-copy | 1338 |
| | 12–14 low-copy | 398 | 12–14 high-copy | 759 |
| | 14–16 low-copy | 85 | 14–16 high-copy | 170 |
| 4-Supplement 2A | 0–2 single-copy | 20,030 | 0–2 multi-copy | 1999 |
| | 2–4 single-copy | 36,494 | 2–4 multi-copy | 2104 |
| | 4–6 single-copy | 34,883 | 4–6 multi-copy | 2098 |
| | 6–8 single-copy | 16,178 | 6–8 multi-copy | 1668 |
| | 8–10 single-copy | 4078 | 8–10 multi-copy | 644 |
| | 10–12 single-copy | 873 | 10–12 multi-copy | 204 |
| | 12–14 single-copy | 215 | 12–14 multi-copy | 66 |
| | 14–16 single-copy | 52 | 14–16 multi-copy | 10 |

*Table 2. Continued on next page*

*Table 2. Continued*

| Figure | Category | n | Category | n |
|---|---|---|---|---|
| 4-Supplement 2B | 0–2 single-copy | 19,892 | 0–2 multi-copy | 1128 |
| | 2–4 single-copy | 36,474 | 2–4 multi-copy | 1111 |
| | 4–6 single-copy | 34,921 | 4–6 multi-copy | 1177 |
| | 6–8 single-copy | 16,208 | 6–8 multi-copy | 1356 |
| | 8–10 single-copy | 4060 | 8–10 multi-copy | 1424 |
| | 10–12 single-copy | 922 | 10–12 multi-copy | 1505 |
| | 12–14 single-copy | 270 | 12–14 multi-copy | 887 |
| | 14–16 single-copy | 56 | 14–16 multi-copy | 199 |

for the reasoning underlying this copy number normalization methodology). The copy number for each bin was determined by splitting the complete genomic sequence into overlapping 20 bp segments at 1 bp intervals and re-mapping to the genome with reads allowed to align to all perfectly matching sequences, producing a measure of the number of genomic sequences matching each 100 bp bin. Little difference was seen if one mismatch was allowed (data not shown).

## Read count normalization for multi-copy sequences

Multi-copy loci are problematic for standard high-throughput sequencing mapping pipelines and are commonly discarded. Reads mapping to a non-unique genome sequence are usually assigned at random to one copy in the genome, therefore the total reads are divided evenly amongst the copies and the apparent abundance of RNA matching each copy is effectively divided by the copy number. In order to assess total RNA abundance (as in *Figure 4A,B*), we multiplied all total RNA read counts by the copy number to obtain the real total RNA abundance. However, we decided that the siRNA abundance should be analyzed per producing locus because each copy in the genome was analyzed separately for comparison with single-copy loci (i.e., we quantified how many siRNA reads an individual copy of a multi-copy locus produced, not how many the combined copies produced). We therefore did not multiply the siRNA read counts by the copy number. Such normalizations clearly have the potential to introduce systematic biases, and we therefore repeated the analysis in *Figure 4* either with no copy number normalization or with both total and siRNA read counts multiplied by copy number (*Figure 4—figure supplement 2*). We found that although the distributions changed somewhat, the majority of total RNA abundance categories had higher siRNA levels for multi-copy loci irrespective of the copy number normalization applied. We note that the alternative copy number analysis in *Figure 4C,D* did not require any normalization; indeed, any copy number normalization would simply cancel out in the calculation of the siRNA:total RNA ratio, therefore any systematic bias that might be introduced by copy number normalization would have no effect.

## Acknowledgements

We thank Simon Andrews, Felix Krueger, Laura Biggins and Anne Segonds-Pichon for Bioinformatics support, Simon Walker for microscopy support, David Bartel for strains, Robin Allshire, Wolf Reik and Sarah Elderkin for comments, and Alex Murray and Tim Hore for critical reading. This work was supported by the Wellcome Trust [grant number 088335].

## Additional information

### Funding

| Funder | Grant reference number | Author |
|---|---|---|
| Wellcome Trust | 088335 | Cristina Cruz, Jonathan Houseley |

The funders had no role in study design, data collection and interpretation, or the decision to submit the work for publication.

## Author contributions

CC, Conception and design, Acquisition of data, Analysis and interpretation of data; JH, Conception and design, Acquisition of data, Analysis and interpretation of data, Drafting or revising the article

## Additional files

### Supplementary files

• Supplementary file 1. Yeast strains, primers, plasmids, and probes used in this study. **Strains**. Strains used in this study. [a]Diploid strains from mating W303α with BY4741. **Primers**. Primers used in this study organized by locus. **Plasmids**. Plasmids used in this study. **Probes**. Probes used in this study for northern and Southern blot analysis. Probe target is given in parentheses after probe name.

### Major dataset

The following previously published datasets were used:

| Author(s) | Year | Dataset title | Dataset ID and/or URL | Database, license, and accessibility information |
|---|---|---|---|---|
| Drinnenberg IA, Fink GR, Bartel DP | 2011 | Compatibility with Killer explains the Rise of RNAi-deficient Fungi | GSE31300; http://www.ncbi.nlm.nih.gov/geo/query/acc.cgi?acc=GSE31300 | Publicly available at GEO (http://www.ncbi.nlm.nih.gov/geo/). |
| Hobson DJ, Wei W, Steinmetz LM, Svejstrup JQ | 2012 | RNA polymerase II collision interrupts convergent transcription (RNA-seq) | GSE38383; http://www.ncbi.nlm.nih.gov/geo/query/acc.cgi?acc=GSE38383 | Publicly available at GEO (http://www.ncbi.nlm.nih.gov/geo/). |

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
