## [Decision Letter]

Thank you for sending your work entitled “Endogenous RNA interference is driven by copy number” for consideration at *eLife*. Your article has been favorably evaluated by a Senior editor and 3 reviewers, one of whom is a member of our Board of Reviewing Editors.

The Reviewing editor and the other reviewers discussed their comments before we reached this decision, and the Reviewing editor has assembled the following comments to help you prepare a revised submission.

This study employs the so-called RNAi+ *S. cerevisiae* strain generated in the Bartel lab (23) that is transformed with *S. Castellii Dicer* and *Ago* genes. This original study suggested that this RNAi+ yeast would be a good model system to explore RNAi gene silencing mechanisms. It is clear that this is the case as this new study makes excellent use of this strain. What is shown here is that endogenous convergent transcription (especially mRNA sense and ncRNA antisense arrangement) generates siRNAs in RNAi+ cells but this does not correlate with reduced mRNA abundance (Figure 1). Also repetitive sequences (Y′ RNA and rRNA) generate abundant siRNAs and show significant RNAi dependent RNA degradation. Some of these data are however predicted from the original Drinnenberg et al study. New results are presented in Figure 3 showing in two test gene cases (*MAL32* and *GAL4*) that forced sense antisense expression of these genes on multicopy transformed plasmids generate both siRNAs and also show clear gene silencing. These results lead to the key observation that it isn't the overall level of expression (sense antisense) that is critical but rather it is the fact that expression comes from multicopy genes (rather than one highly expressed gene). This point is initially made from bioinformatic analysis of the RNAi+ transcriptiome (Drinneneberg et al) where multicopy genes generate more siRNA than single copy rather than simply correlating with totals RNA levels (Figure 4). Figures 5 and 6 then validate the bioinformatics very clearly. The integrated single copy *MAL32* expressing high level sense and antisense transcription due to the use of strong convergent arranged promoters does not generate as much siRNA as multicopy low expressed *MAL32*. Also sense antisense ncRNAs expressed on multicopy plasmids generate significant amounts of siRNA. Finally some evidence was provided that siRNAs from high-copy *GAL10* constructs (making sense and antisense RNAs) can generate siRNA that silence another *GAL10* gene in trans.

Overall the three reviewers of this paper while aware of the somewhat artificial nature of the RNAi+ *S. cerevisiae* strain consider that the experimental benefit of this system has been put to excellent use and has allowed significant new insight in RNAi mechanism to be obtained. The key additional experiments that we feel are necessary to make a revised manuscript acceptable for publication by *eLife* are as follows.

1) The actual copy number of the 2µ plasmids transformed into the RNAi+ strain should be directly measured by Southern blotting

2) The *Castellii* Dcr1 gene should be knocked out to show loss of the high-copy number silencing observed for *MAL32* and *GAL4*.

3) The location of dsRNA and *Dicer* in the transformed RNAi+ cells should be directly visualised (J2 antibody for dsRNA and Tagged *Dcr1*/anti *Dcr1*) to show either chromatin or cytolasmic association. See Schönborn, J. et al. NAR 19, 2993-3000, 1991; Gullerova and Proudfoot NSMB 19, 1193-1201 2013 for J2 antibody.

4) Better quantitation of RNAi effect on mRNA levels and level of dsRNA is required. This can be achieved by RTqPCR and for dsRNA level of J2 dsRNA immunoprecipitation.

---

## [Author Response]

*1) The actual copy number of the 2µ plasmids transformed into the RNAi+ strain should be directly measured by*
*Southern blotting*

We have included a southern for 2µ copy number as Figure 3—figure supplement 5. We have also shown a northern blot for the *MAL32* high-copy plasmid in which the plasmid signals are clearly visible. Both analyses show that the copy number does not decrease in the RNAi+ strain – it actually increases, an observation we discuss in the text.

*2) The* Castellii *Dcr1 gene should be knocked out to show loss of the high-copy number silencing observed for* MAL32 *and* GAL4*.*

We decided to combine this with changing the genetic background of the RNAi+ strain. We cloned Dcr1 and Ago1 onto single copy plasmids and introduced these into BY4741 cells (the original background is W303), along with the high-copy plasmids. The result for *MAL32* is very clear, showing that both genes are required for knockdown (Figure 3—figure supplement 3). However, in this genetic background *GAL4* is only expressed very weakly from the high-copy plasmid; the RNA forms dsRNA efficiently as expected and this is removed by Dicer, although this does not lead to a clear knockdown. The *GAL4* knockdown was the weakest we observed in the original experiments, and RNAi knockdowns in this new strain are not as strong as in the original strain (60% in new strain vs. 80% in the original strain for the *MAL32* mRNA). Combined with the confounding effect of the increased plasmid copy number in RNAi+ cells, the *GAL4* knockdown becomes too weak to test the importance of Dicer. We do not think that this compromises the conclusions of the paper.

*3) The location of dsRNA and* Dicer *in the transformed RNAi+ cells should be directly visualised (J2 antibody for dsRNA and Tagged* Dcr1*/anti* Dcr1*) to show either chromatin or cytolasmic association. See Schönborn, J. et al. NAR 19, 2993-3000, 1991; Gullerova and Proudfoot NSMB 19, 1193-1201 2013 for J2 antibody*.

We have included this data as Figure 6—figure supplement 1. We could not convincingly detect Dicer in nuclei, and the dsRNA signal is dominated by strong cytoplasmic signals, most likely from Killer virus RNAs. Although RNAi should technically clear Killer virus, the analysis performed by the Bartel lab shows that this clearance is far from complete. We have used a different approach to improve confidence in nuclear formation of dsRNA; if dsRNA forms in the nucleus then the juxtaposition of transcribing copies should be important, and we demonstrate that this is true for the *TRP1* gene in the new Figure 6. We have referred to both of these experiments in the Discussion.

*4) Better quantitation of RNAi effect on mRNA levels and level of dsRNA is required. This can be achieved by RTqPCR and for dsRNA level of J2 dsRNA immunoprecipitation*.

We have added a number of additional results here. Firstly, we have added quantification for the effect of RNAi on endogenous *MAL32* and *GAL4* mRNA along with full length northern blot images that show degradation patterns (Figure 3—figure supplement 2). We have also performed the J2 immunoprecipitations (Figure 3—figure supplement 6) – these show a clear loss dsRNA in the RNAi strain, and confirm the Northern blot results for mRNA knockdown.